# Implementation of the Infant-Toddler Checklist in Swedish child health services at 18 months: an observational study

Anton Dahlberg, Anna Levin, Anna Erica Fäldt 

Child Health and Parenting (CHAP), Department of Public Health and Caring Sciences, Uppsala University, Uppsala, Sweden

**Correspondence to**
Dr Anna Erica Fäldt; anna.faldt@uu.se

## ABSTRACT

**Background** Communication and language disorders are common conditions that emerge early and negatively impact quality of life across the life course. Early identification may be facilitated using a validated screening tool such as the Infant-Toddler Checklist (ITC). We introduced the ITC at the 18-month visit to child health services (CHS) in a Swedish county. Using the RE-AIM implementation framework, this study assessed the implementation of the ITC according to five key dimensions: reach, effectiveness, adoption, implementation and maintenance.

**Methods** This observational study used medical records at CHS as data source. Data were collected from children who visited a child health nurse at 17–22 months. The sample included 2633 children with a mean age of 17.8 months, 1717 in the pre-implementation group and 916 in the post implementation group. We calculated the ITC completion rate (reach) and use at each site (adoption). We compared rates of referral to speech and language therapy (effectiveness) before and after implementation of the ITC using OR and 95% CIs. We described actions to facilitate implementation and maintenance of ITC screening over time.

**Results** The overall screening rate was 93% (reach) which increased from 80% initially to 94% at the end of the 2-year period (maintenance). All centres used the ITC (adoption). The ITC screen positive rate was 14%. Of children who had reached at least 24 months (n=2367), referral rate was 0.4% pre-implementation versus 6.9% post implementation (OR=18.17, 95% CI 8.15, 40.51, p<0.001) (effectiveness). Implementation strategies included training sessions, collaboration, written and automatic procedures and modifications to the medical records system.

**Conclusion** The implementation of the ITC was associated with high reach, higher referral rate, complete adoption, and sustained maintenance over time.

## INTRODUCTION

Communication and language are the basis of human interaction and are essential for development, learning and health. Language and communication difficulties can affect a child's whole life and lead to a lower quality of life, including mental health problems, school difficulties and social exclusion.[1 2]

### WHAT IS ALREADY KNOWN ON THIS TOPIC
⇒ Early language assessment is essential for offering timely interventions to children with communication disorders.
⇒ Implementing and maintaining new screening procedures in healthcare is difficult and research on this is scarce.

### WHAT THIS STUDY ADDS
⇒ The Infant-Toddler Checklist (ITC) can be implemented relatively quickly.
⇒ After a short implementation period and adaptation of procedures, ITC was used in all units and almost all children were screened with ITC. ITC can be an effective instrument for identifying children earlier, thus enabling timely interventions.

### HOW THIS STUDY MIGHT AFFECT RESEARCH, PRACTICE OR POLICY
⇒ Stakeholders at the child health services should consider implementing a scientifically evaluated screening method at 18 months to enable early interventions in settings where interventions are available.

Language and communication difficulties are common affecting around 10% of children in the Western world.[3] There is also considerable overlap between language and communication difficulties and other neurodevelopmental conditions.[3–5]

Early interventions for language and communication difficulties affect children's language development and are hypothesised to reduce the risk of secondary negative impacts such as disturbance in emotional and social adaptation.[6–10] To enable timely interventions, there is a need for validated and evidence-based screening instruments.[11] Several methods are available to identify children with communication and language difficulties early,[12 13] but implementation in child health services (CHS) is still in its infancy. The Swedish CHS is regulated by a national programme[14] and reaches almost all children, aiming to promote health and monitor

children's development,[15] making them an excellent arena for early language screening. Children's communication and language are monitored at 18 months by asking parents if the child speaks at least 8–10 words and understands more than 8–10 words. This practice has been found to have a low sensitivity[16]; thus, more accurate screening methods are needed.

One screening instrument for identifying children with language and communication difficulties is the Communication and Symbolic Behaviour Scales Developmental Profile Infant-Toddler Checklist (ITC).[17] [18] The ITC detects a range of developmental concerns, including language delay, global developmental delay and autism.[13] [19] [20] When used at the 18-month visit in Canadian primary care, the ITC had high specificity (92%) and negative predictive value (96%), and a low sensitivity (31%) and false positive rate (8%) for a developmental diagnosis at 3–5 years.[21] ITC used at the 18-month visits in a Swedish CHS showed a sensitivity of 85% and specificity of 59% which increased to 88% and 63%, respectively when the ITC was combined with the nurse's assessment.[22] Further, nurses have highlighted the usability and relative advantages of using the ITC in CHS.[23]

Introducing new approaches[24] and transferring methods tested in research studies into everyday clinical practice is complex and takes time.[25] Applying an implementation framework when assessing an intervention allows one to investigate the process's *how* and *why*, which can facilitate identifying and understanding effective components and barriers in the implementation process.[26] This, in turn, can form a rationale for future implementation.

In this study, the RE-AIM framework was applied to describe the implementation of the ITC in CHS.[27] The RE-AIM offers a structured way of describing, employing and analysing implementation processes at five key dimensions: the *reach* of the target population, *effectiveness*, *adoption*, *implementation* and *maintenance* of the implemented method.[25]

### Research questions

1. What proportion of eligible children were screened with the ITC after its implementation? (Reach)
2. Did the implementation of the ITC lead to earlier referrals to speech and language therapy (SLT) for children with suspected language or communication difficulties? Were there any sex differences in referral rates? (Effectiveness)
3. Was the ITC adopted equally across all three centres? (Adoption)
4. What actions were taken to facilitate the implementation of the ITC during the study period? (Implementation)
5. Was the implementation of the ITC sustained over time, and if so, how? (Maintenance)

## METHODS

### Study design

The study design used an observational study design based on medical records. As this study was based on data from medical records, involvement of patients was not possible.

### Screening instrument

The ITC consists of 24 multiple-choice questions completed by the parents. It is divided in three composite scores: social, speech and symbolic, and a total score.[17] [18] If the social or symbolic subscales or the total score are below a cut-off value of the 10th percentile (concern) and thus indicate a possible divergent communication development, the child should be referred to a specialist. If only the speech scale indicates concern, the child should be reassessed with the ITC after 3 months.[13] If the ITC indicates concern about the speech composite on two occasions the child should be referred.

### Data collection

Regional documentation regarding implementation strategies was collected from the CHS unit. Pseudonymised data, including child age at screening, sex, screening method and results, referral and age at referral to SLT and centre, were collected via the medical records system.

As the ITC-specific item-selection keywords were not applied in the medical records system during the first months after the implementation, both data based on free text entries and ITC-specific item-selections were processed. The collected ITC free-form text entries contained information on completing the ITC screening (*ITC completed/ITC not completed*) and potential concern for the child's communication development (*concern/no concern*). The concern data was collected through record entries, and it was not specified whether they were based on the ITC *concern* item or on parent interviews during the visit. Stated reasons for not referring children with an indication of concern about failing the screening, were collected.

### Setting and population

The ITC 18-month screening was introduced in the Region of Gotland, the smallest county in Sweden with 500–600 children born annually. The CHS is divided into three centres in different geographical areas with different socioeconomic demographics.[28] The nurses were supported in implementation, consultation and education by a central CHS unit consisting of a nurse, a paediatrician and a psychologist.

The ITC was introduced for children born in January 2019 and onwards. Due to the summer vacation period in Sweden, three children scheduled for July visits had their 18-month visits in June 2020. However, July was considered the starting point for assessing reach and maintenance.

**Table 1** Characteristics of the participants in the two samples

| | n (%) | |
|---|---|---|
| **Full sample** | **Care as usual** | **ITC** |
| | n=1717 | n=916 |
| Sex | | |
| Girls | 838 (48.8) | 446 (48.7) |
| Boys | 879 (51.2) | 470 (51.3) |
| Concern | | 70 (7.6) |
| Speech | | 50 (5.5) |
| Subsample including children aged at least 24 months or referred early | | |
| | n=1717 | n=650 |
| Sex | | |
| Girls | 838 (48.8) | 314 (48.3) |
| Boys | 879 (51.2) | 336 (51.7) |
| Concern | | 58 (8.9) |
| Speech | | 35 (5.4) |

ITC, Infant-Toddler Checklist.

To answer the different research questions, two samples were used (table 1). The first sample, answering research questions 1 and 3–5, included all children receiving their 18-month CHS visit (17–22 months) between May 2016 and April 2022 (n=2633 children, girls 48.8%). Mean age at visit was 17.8 months (SD=0.93). The intervention group (n=916, 48.7% girls), henceforth the ITC group, was exposed to ITC (June 2020–April 2022), while the control group (n=1717, 48.8% girls) received care as usual (May 2016–June 2020).

To answer research question 2, regarding the effectiveness of the ITC in terms of early referrals (before 25 months of age), a sample was used including children who had reached at least 24 months of age or had been referred to a speech and language therapist before turning 24 months (n=2367). This was based on clinical experience and previous studies showing that it can take time to motivate parents to refer after screening with concern at 18 months.[23] The age limit also allowed for repeated screenings when appropriate. The control group consisted of 1717 children (48.8% girls), and the ITC group of 650 children (48.3% girls).

## Analysis

Reach was assessed through the ratio between the screened children and the total number of children in the ITC group. The effectiveness was examined through descriptive statistics with the number of children where the ITC result indicated concern, and referrals. To analyse the differences in early referral rates (effectiveness), the $\chi^2$ test was used. Effect size was measured through OR and 95% CI, using logistic regression. Sex differences in early referral were assessed through $\chi^2$ test. The alpha level for the $\chi^2$ analyses were set to 0.05. Data were prepared and analysed using the statistical software R (V.4.2.2) and SPSS (V.28).[29 30] Adoption rates were analysed using descriptive statistics. Implementation was assessed qualitatively through written documentation and information from staff within the central CHS unit responsible for the implementation process. Maintenance was assessed using descriptive statistics.

## RESULTS

Following the structure of the implementation framework, results are presented for each domain (table 2).

## Reach

Of the 916 children in the ITC group, 854 were screened with the ITC, representing a reach of 93%. For 11 children, the ITC was reported as failing to be returned to the CHS. For the remaining 51 children, no reasons were given for the children not being screened. See figure 1 for a flowchart of screening results and referrals for girls and boys.

**Table 2** Objectives and results for each of the RE-AIM domains

| | **Reach** | **Effectiveness** | **Adoption** | **Implementation** | **Maintenance** |
|---|---|---|---|---|---|
| Objective | Proportion of eligible children screened with the ITC after its implementation | Effect of the ITC implementation on earlier referrals for children with suspected language or communication difficulties | Equality of adoption across CHCs | Actions taken to facilitate the implementation of the ITC | If and how the implementation of the ITC was sustained over time |
| Result | 93% screening rate (854/916 children) | Significant difference in referrals between ITC and previous method (OR=2.07, 95% CI 1.34, 3.14). Being a boy were associated with early referral (OR=2.38, 95% CI 1.30, 4.36, p<0.001) | All centres adopted the method (100% adoption rate) | ► Training session<br>► Close collaboration<br>► Written procedure<br>► Automated procedure<br>► Modified medical record procedure | Time to full implementation: 3 months Implementation maintained over full study period Brief drop in screening rate during first 6 months |

CHS, child health services; ITC, Infant-Toddler Checklist.

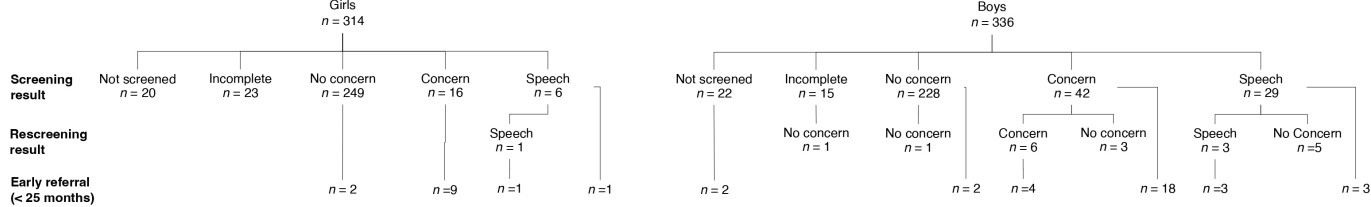

**Figure 1** Flowchart of screening results and referrals for girls and boys.

## Effectiveness

For this RE-AIM domain, the subsample was used. Of the 608 children screened with the ITC, 58 children (7.9%) had a result indicating concern. Additionally, 35 (6.3%) children had concern on the speech composite only and thus would be rescreened with the ITC 3 months later. However, four children were referred after the first screening. Of the remaining 31 children, 22 were not rescreened.

For 20 children screening with the ITC had been performed on two occasions. Nine of these children had concern on the speech composite only. The rescreening indicates concern for 10 children whereof four in the speech composite. In total 24 children with concern the ITC were not referred. For nine of these parents expressed no concern or declined the referral offer. The nurses stated no concerns regarding the child's communication development for six. Of the remaining nine children no reasons were given. Additionally, 22 children were not referred or only screened with ITC once after a positive screening on the speech composite.

Analysing the subsample of children aged at least 24 months or referred early, a significant relationship between screening method and early referral was detected, where the ITC was associated with more early referrals ($\chi^2$ (n=2367)=93.2, p<0.001). Referral rates were 0.4% in the care as usual group and 6.9% in the ITC group. The odds for early referral were substantially higher for children screened with the ITC (OR=18.17, 95% CI 8.15, 40.51, p<0.001). Further, being a boy was associated with an earlier referral (OR=2.38, 95% CI 1.30, 4.36, p<0.001). In the control group, 28.6% of the referred children were girls, while this rate was 28.9% in the ITC group. These rates did not significantly differ from each other.

## Adoption

All centres implemented ITC, resulting in a 100% adoption rate at setting level, which remained unchanged during the study period. At the centre level, there was variation in the proportion of children screened (range 92%–97%) with the highest adherence in the most socioeconomically challenged area.

## Implementation and implementation strategies

The implementation was performed in collaboration between the central CHS unit and the SLT services. Prior to the implementation of the ITC, staff attended training sessions and workshops on parent–child relationships, early communication development, the importance of early intervention and how to perform the ITC. A written procedure was developed, including referrals and re-screenings. Shortly after implementing the ITC, a medical record system was modified to include an ITC-specific template with fixed response options for record entry. Before the modification, the nurses entered the results of the ITC screening in free-form text fields into the medical record. Further, the medical record system automatically included the ITC with the invitation letter to the 18-month visit.

## Maintenance

During the study, the CHS unit, mainly through a designated project manager, kept close contact with the nurses, offering support, feedback and monitoring the implementation process. The screening procedure instructions were continuously updated and improved throughout the implementation. The use of the ITC was sustained over time during the entire study period and is, as of 2023, still part of routine practice across the county.

The consistency of implementation varied in the early phase of implementation, as the proportion of children screened was initially low: in the first 3 months, 20% of the children were not screened. Thereafter, the rate of screened children varied between 94% and 100% until January 2022 (figure 2). During this month, the rate of screened children dropped to 87%. This coincided with the start of a short-term employment of a substitute nurse. By March 2022, the rate of screened children had increased to 94%.

## DISCUSSION

Introducing new procedures and methods to everyday clinical practice is often complicated and challenging.[31] It is, therefore, essential to monitor the implementation processes. This study aimed to enhance knowledge regarding implementing the ITC through the RE-AIM framework.[25 27] A vast majority of the children were screened with the ITC after its introduction (93%). More children were referred early to speech and language assessment after the introduction of the ITC compared with care as usual. The overall adoption rate was 100%, with all three centres adopting the method from its introduction. Strategies for successful implementation included a training session, close collaboration with external parties, written and automated procedures, and

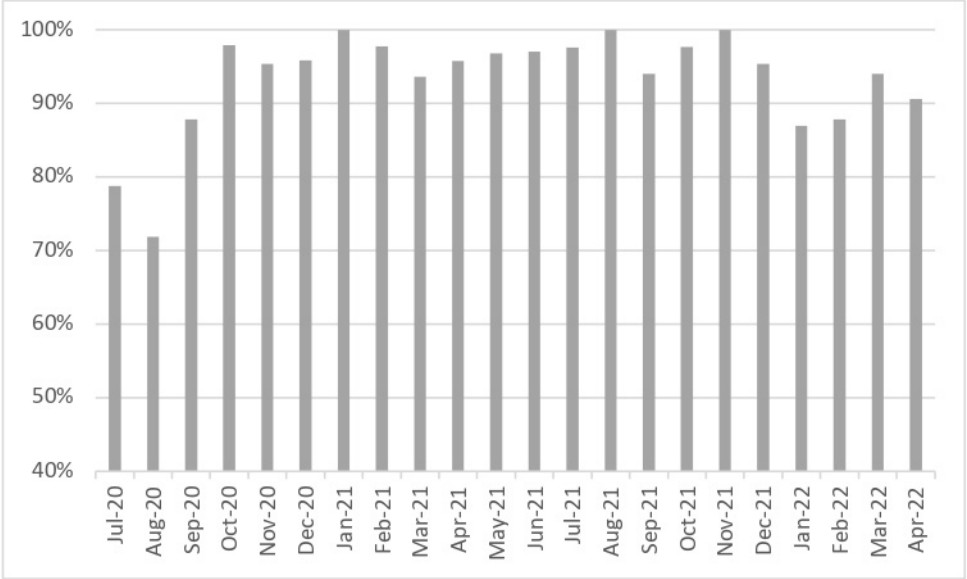

**Figure 2** Screening rates across implementation time.

modified procedures regarding medical record keeping. The implementation maintenance was sustained over time.

The reach of the ITC was high; 93% of children who visited the CHS since the introduction were screened with the ITC. One risk of using screening instruments is that children are not identified equally across demographics.[32] No individual sociodemographic data were available in this study. Therefore, it is impossible to conclude whether children are screened and identified equally. However, the three centres were located in areas with diverse socioeconomic makeup, and results suggested that adoption was highest in the most socio-economically challenged areas.[28]

There was a large difference in referrals between girls and boys with an OR at 2.38, using both usual care and the ITC, although the difference tended to be smaller after the introduction of ITC. Language and communication difficulties have been shown to be more common in boys than in girls[33] and a recent study found an adjusted OR at 2.15 for boys using ITC.[34] Other studies have found that the sex differences are often minor and non-significant.[3] On the other hand, the boy–girl ratio for receiving intervention is significant,[35] which can result from differences in identification. It is essential to study reasons for sex differences in identification and referral in future studies.

Implementation took place with inter-organisational and inter-professional collaboration. This presumably positively affected the implementation and facilitated planning and communication in the short and long term. As anticipated, the implementation had a start-up period of 3 months before almost all children were screened with the ITC, a period considered short compared with other implementations.[36] Preparations for the implementation included workshops and training for the nurses. This can be seen as a beneficial factor to the high adoption rate

and reach and an important part of the implementation, as the nurses were relatively well prepared for the implementation process.

The central CHS unit included a designated project manager, who followed the implementation of the ITC closely. The continuous improvement of the written procedures and routines may have facilitated the implementation, created reassurance for the nurses and reduced the risk of incorrect practice. The automated printing and including the ITC with the invitation letter to the parents reduced the risk of failing to send out the questionnaires. Further, the revised medical journal template reminded the nurses and facilitated correct record-keeping.

The decrease in the proportion of screened children during staff turnover may indicate the importance of ongoing training and supervision to facilitate maintenance of screening. However, since the number of children having health visits each month was relatively low, one needs to exercise caution when interpreting (figure 2).

Not all children who should have been rescreened after concern on the speech composite were rescreened and some children who had an ITC result indicating concern were not referred to SLT. While this was mainly explained through parents declining referral or the nurses' clinical assessment for some children, no reason was given for not sending a referral for nine of the children. Adherence to referral guidelines can be low and might depend on family demographics.[37] Even though ITC has been described as giving parents insight into their child's development and facilitated referrals, it can be difficult to motivate parents referral to SLT at 18 months.[23] Professionals sometimes hold off on referral, even when parents express concern.[38] The psychometric properties of the ITC in the Swedish setting with a somewhat low specificity could be a possible contributing factor

to the lower referral rate.[22] Other studies have shown a high specificity and low sensitivity of ITC screening at 18 months.[21] However, methodological differences may account for the discrepancies to some extent. Problems with specificity and sensitivity highlight the need for repeated screening and health surveillance. Screening may need to be supplemented with the nurse's clinical observation and a dialogue with the parent.

The goal of implementing the ITC was to ensure that children needing support in their communication and language development would be identified earlier and referred to specialist services such as SLT. For early referrals to be meaningful, interventions must be available in conjunction with the referrals. In the Region of Gotland, such a timely intervention is available, which is not the case in all of Sweden, where the waiting times for SLT interventions can be years long.[39]

One basic principle of screening is that only children needing interventions should be identified,[40] which is especially important when resources are scarce. This puts extra demands on the use of clinically valid screening instruments. The ITC could be one such instrument, as the analyses show that the referral rate was equal before and after the implementation of the ITC but that the referrals occurred when the children were younger. Future studies should assess how ITC affects the care process for children how the ITC relates to later language screenings.

## CONCLUSIONS

The implementation of the ITC was characterised by a high reach, higher rates of early referrals, complete adoption and sustained maintenance over time. Thus, the ITC implementation process was successful in relation to the RE-AIM domains. The successful implementation was facilitated through training, improvements in the medical record system, close monitoring of the implementation and written procedures that may have contributed to the outcome.

The ITC can be used for early detection of language and communication difficulties and hence help families receive timely interventions. Depending on context, a similar implementation could be beneficial, as there is a general need for earlier screening of language and communication difficulties to enable timely interventions.

**Acknowledgements** The authors would like to thank the child health service in the Region of Gotland for their contribution.

**Contributors** AEF, AD, AL: conceptualised, designed the study and drafted the original manuscript. AEF and AD: data analysis, AL: performed electronic health record data extraction. All authors reviewed and approved the final manuscript as submitted. AEF willingly accepted the role of guarantor for this study.

**Funding** The study was funded by Länsförsäkringar Gotland.

**Competing interests** AL, was involved in the planning and implementing of the ITC in the Region of Gotland.

**Patient and public involvement** Patients and/or the public were not involved in the design, or conduct, or reporting, or dissemination plans of this research.

**Patient consent for publication** Not applicable.

**Ethics approval** This study was conducted according to the ethical guidelines described in the Declaration of Helsinki and was approved by the Swedish Ethical Review Board (Dnr 2021-04737). As the study was based on anonymised previously collected medical record data no consent was collected.

**Provenance and peer review** Not commissioned; externally peer reviewed.

**Data availability statement** Data are available upon reasonable request. The datasets used and/or analysed during the current study are available from the corresponding author on reasonable request.

**ORCID iD**
Anna Erica Fäldt http://orcid.org/0000-0001-7257-8758

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
