## [Reviewer comments · BMJ Paediatrics Open]

ARTICLE DETAILS

TITLE (PROVISIONAL)	Implementation of the Infant-Toddler Checklist in Swedish child health services at 18 months: an observational study
AUTHORS	Dahlberg, Anton Levin, Anna Fäldt, Anna Erica

VERSION 1 – REVIEW

REVIEWER	Ms. Kimberly M. Nurse University of Toronto, Institute of Health Policy, Management and Evaluation
REVIEW RETURNED	21-Dec-2023

GENERAL COMMENTS	This study used the RE-AIM implementation framework to assess whether children with language and communication concerns at 18 months were identified and referred earlier after the introduction of the ITC compared to usual care. Strengths of this study include that it was conducted through Swedish child health services and the ITCs implementation was well-supported by health care providers. Ensuring early identification of communication and language delays using validated screening tools as well as referral and access to timely interventions is an important goal. Research Questions:  - Consider reframing the research questions as specific objectives or aims that correspond with the key dimensions of the RE-AIM framework. Methods:  - Include that the ITC is completed by a parent or caregiver. - Wetherby and Prizant (the ITC developers) refer to three composite scores and a total score, the composite score terminology is recommended instead of subscales (please see the reference below). - Reference: Wetherby, Amy & Prizant, Barry. (2001). Communication and Symbolic Behavior Scales Developmental Profile Infant/Toddler Checklist. https://www.researchgate.net/publication/252234671_Communication_and_Symbolic_Behavior_Scales_Developmental_Profile_InfantToddler_Checklist - In line 113, instead of the use of the word “fail”, indicate what the cut-off value is (the 10th percentile). - It may be beneficial to include “child age at screening, sex, screening method and results, referral and age at referral to SLT and centre” (referenced in lines 118 & 119) in a table of participant characteristics in the Results (as Table 1). - It may be useful to emphasize that informed parent consent was obtained as part of ethics.
--

	 - Under study population (line 129) instead of subsample, this seems to be the total sample and a subsample of this was used to determine effectiveness as everyone in the intervention group had not yet reached 30 months of age. It would be beneficial to update the text to refer to the total sample (line 129) followed by the subsample noted on line 134. - Please include a rationale for using 25 months as the cut-point for early referral. The above Wetherby and Prizant (2001) paper may be referenced for this. - Please provide more details around the type of natural experiment used. - Throughout the paper, it is recommended to use the concern terminology used by Wetherby and Prizant rather than "fail". - Fisher's exact test is typically used to address small cell counts. Please include a brief rationale for the use of Yates' continuity correction. - Please clarify the methods used to address the research questions related to the AIM domains. - It would be beneficial to include the version of R and SPSS used for this analysis. - It would be beneficial to include the alpha level (e.g., 0.05) and whether statistical tests were 2-sided in the methods section. Results:  - Lines 146 & 147 (Figure 1): Include a title for Figure 1; rephrase the research questions as objectives (as indicated above); for Reach, include the ratio as well as the percentage; and for Effectiveness, include the confidence interval. - Where possible, it may be beneficial to display study results detailed in the text in tables and figures as well. - Under Effectiveness:  o Please clarify whether the 18 children with scores in the concern range on two occasions had speech composite scores in the concern range. o Please clarify whether the following population was captured in this study: children with a speech composite score of concern, who were reassessed after three months and referred to a specialist if the concern remained. o For the children whose parents expressed no concern, please comment on whether this was based on the ITCs open question on parental concern regarding child development. o There seems to be a slight calculation error: Of the 75 children with scores in the concern range who were not referred early to SLT, only 74 are accounted for in the text. Discussion:  - In lines 221 & 222, reference is made to children who "screened positive with the ITC". A positive ITC screen should be explained in the sections above.
--	---

REVIEWER	Dr. Patricia Parkin Hospital for Sick Children, Paediatrics
REVIEW RETURNED	01-Jan-2024

GENERAL COMMENTS	Thank you for the opportunity to review this manuscript which was of great interest since our research group has also been studying the ITC at the 18 month visit in Canadian primary care settings (reference 23). We agree with the authors that the ITC is a promising screening tool for early detection of children with a range of developmental concerns, including language delay, global
---

	developmental delay and autism spectrum disorder. Therefore, I am very enthusiastic about this manuscript. Due to my interest, I read the manuscript carefully and my review is lengthy. However, I think the authors will be able to respond easily to my comments. My most substantial suggestion is to add Tables and Figures to provide greater detail and clarity to the findings. Below are some line-by-line comments. Line 68-70 However, screening for language disorder is not recommended for children younger than four.¹¹ 12 In contrast, early screening is recommended for autism, where language or communication difficulties can be the first symptom.¹¹ 13 – For readers not familiar with the details of these reviews/recommendations this may be unclear. My interpretation of the current recommendations is that evidence for the effectiveness of screening is weak and screening tools demonstrate poor accuracy. The reviews by Jullien (references 11 and 13) are important, but it is notable that the sources (eg USPSTF, NICE) do not include the most current literature. The authors of the present article may want to modify these sentences. Line 78 One method for identifying children... - The authors may want to emphasize 'screening', for example: One screening instrument for identifying children... Line 80-82 When the ITC was used in primary care children with developmental delay were identified with high specificity, high negative predictive value and low false positive rate.²³ – Reference 23 is from our research group. May I suggest a modification to this sentence: When used at the 18 month visit in Canadian primary care, the ITC had high specificity (92%) and negative predictive value (96%), and a low sensitivity (31%) and false positive rate (8%) for a developmental diagnosis at 3–5 years.²³ Line 82-84 ITC used at the 18-month visits in a Swedish CHS showed a sensitivity of 0.85 and specificity 0.59 which increased to 0.88 and 0.63, respectively when the ITC was combined with the nurse's assessment.²⁴ – I have reviewed reference 24 from the same authors. Readers may wonder why the sensitivity and specificity from the two studies (references 23 and 24) are so different. This could be elaborated on in the Discussion section. In reference 24, the authors state in limitations: "Not all children with a positive screen were referred and were therefore not assessed". In our study, criterion measures were available for all children with positive or negative screening. This may account for the differences in study findings. Line 86-91 Setting – I think this paragraph can be moved to the METHODS section.
--	---

	Line 92 Implementation strategies – as this paragraph is part of the INTRODUCTION section, I don't think this subheading is needed and could be removed. Line 102-109 Research questions – This paragraph is well written, clear and much appreciated. The term for each of the 5 dimensions in brackets at the end is very helpful. Line 106 3. Was the ITC adopted equally across all centres? (Adoption) – Am I correct in understanding that there were 3 centres? Could the number of centres be included here? Line 104 2. Did the implementation of the ITC lead to earlier referrals...The authors collect child sex and report on differences in early referrals in Results. Would they consider adding this to research question #2? Line 110 METHODS – the authors could consider adding as the first subheading: Study Design. On Line 136 the authors describe their study as a 'natural experiment' and provide reference 31. I think they could consider identifying their study design as an observational study design using data collected in medical records as the data source. Line 110 METHODS – After Study Design, the authors could consider a subheading for Setting and Population. As currently written, I found these sections (Setting lines 86-91 and Population lines 128-135) somewhat confusing. For example, the Setting section notes that ITC was introduced in January 2019; but the Population section notes that usual care was received 2016-2020 and ITC was received 2020-2022. Would the authors consider merging these 2 sections and clarify the setting and population? Line 128-137 Study population – this section is very helpful. The second paragraph describes the study population for research question #2 (effectiveness). Can the authors clarify if the first paragraph describes the study population for research questions #1, 3-5? The word 'subsample' is used – does this mean that this was a subsample of the total sample or is it more accurate to write that these research questions were answered using the total sample? Would the authors consider modifying the first two sentences such as: "To answer the different research questions, two samples were used. The first sample addressed research questions 1,3-5 and included all children..." Line 114-115 If only the speech scale indicates concern, the child should be reassessed with the ITC after three months.16 – Is it correct that if there is concern on the speech composite on two occasions, then the child should be referred for SLT? If so, could this be added? Line 136 As this study was based on register data, no involvement of patients were possible – Does the journal require a specific
--	--

	subheading “Patient and public involvement”? By using the term ‘register data’ are the authors referring to data collected from medical records? If so, perhaps they can use this terminology (medical records) to be consistent with the terminology used earlier in the manuscript. Line 138-144 Analysis – The authors describe the analysis plan for research questions #1 and #2. However, I don’t think they described the analysis plan for research questions #3-5. Could this be added? Line 139 and 140 Reach was explored...The effectiveness was explored... - Some methodologists/statisticians prefer to use the word ‘explore’ in specific circumstances. Would the authors consider using the word ‘examined’ or ‘assessed’ rather than ‘explored’? Line 142 Effect size was measured through odds ratio – Were 95% confidence intervals calculated? Could these be added? Line 142 Gender differences... - In Data Collection the authors used the term ‘sex’ which may be more appropriate on Line 142 as well as Figure 1 and Line 204. Line 145 RESULTS – is it possible for the authors to create a Flow Diagram Figure as per STROBE? Line 145 RESULTS – Could the authors create a Table 1 Participant Characteristics at screening (n=2633) according to two groups (usual care or ITC) including child age, sex, results of screening (concern for social, speech, and symbolic, and total score), referral (yes/no), age at referral to SLT. These are the variables described in Methods-Data Collection section. Line 146 Figure 1 – this is a very helpful summary. Could this be labeled a Table rather than a Figure? Line 152 Effectiveness – this important section would be strengthened with an accompanying Flow Diagram. Line 153-154 ... of which 51 children failed on speech – could the authors add the %, ie “of which 51 (42.5%) children failed on speech Line 154 ...and thus would be rescreened with the ITC three months later – to clarify, were all 51 rescreened three months later? Or were some missed? These data could be added to a Flow Diagram for clarity. Line 154-155 For 18 children, screening with the ITC had been performed on two occasions, failing on both occasions – to clarify, is this 18 of 51 (35%) who failed on speech? Does this number (18
--	---

of 51) refer to the number who were screened twice (ie 33 of 51 did not receive a second screening)? Or does this mean that all 51 received a second screening and 18 of the 51 (35%) failed both times and were eligible for referral? A Flow Diagram would help clarify.

Lines 155-160 – These sentences are valuable but the interpretation would be strengthened by a Flow Diagram and including %.

Line 161 Analysing the subsample of children aged at least 30 months... - this section is very important. Could the authors create a Table 2 of the subsample of children aged at least 30 months (n=2163) according to usual care (n=1717) or ITC (n=446) with similar variables to Table 1 and a column for OR and 95% CI? In the abstract, the authors present the mean referral age in the usual care vs ITC group; however, this data is not included in the manuscript.

Lines 164-166 Sex differences in early referrals could be reported in Table 2 with %, OR and 95% CI.

Line 180 Maintenance – this section would be strengthened with a Figure showing screening rates over time. Methods of assessing and analyzing maintenance were not described in Methods and could be added.

Line 190 DISCUSSION – could the authors summarize their findings in the opening paragraph with a sentence for each of the 5 research questions?

Line 200 There was a large difference in referrals between girls and boys using both methods... - could the authors specify the direction and magnitude of the differences (eg two times higher for boys)? And specify the meaning of 'both methods' (ie usual care or ITC screening).

Line 201 Language and communication difficulties have been presumed to be more common in boys than in girls – Using the word 'presumed' suggests that there is no literature on this. Can the authors clarify whether there is research evidence to support this? Our research group is also interested in factors associated with positive ITC. We recently reported: The aOR for a positive ITC for male compared with female sex was 2.15 (95% CI, 1.63-2.83; P < .001). (J Pediatr 2024;264:113769).

Line 222 While this was mainly explained through best practice... - can the authors clarify the meaning of this? Are they referring to children who screen positive on the speech composite on first screening? I was unclear in Results section how many of these children underwent a second screening and of these how many screened positive again and were subsequently referred to SLT.

Line 228 ...as the specificity of the ITC is somewhat low when used at 18 months.²⁴ This is a relatively common challenge in screening at younger ages – This is a very interesting interpretation according to the authors previous research (reference 24). In contrast, our research group has found screening at 18 months in a low risk primary care population to have high specificity (92%) and negative predictive value (96%), and a low sensitivity (31%) and false positive rate (8%) for later developmental outcomes (reference 23 as well as *Academic Pediatrics* 2023;23:322–328). We have interpreted the findings of low sensitivity from our ITC studies as well as studies of other screening tools at younger ages as highlighting the importance of ongoing developmental surveillance and screening beyond 18 months.

Line 243 The implementation of the ITC was associated with high reach, earlier referrals, complete adoption, and sustained maintenance over time. – This is an excellent summary sentence and a similar sentence would be used for the opening paragraph of Discussion with elaboration on each of the dimensions.

Conclusion - I note that for effectiveness, the authors state ‘earlier referrals’, however a comparison of mean age in the two groups is not included in the Methods or Results (but is noted in the Abstract). Was the primary outcome of effectiveness age at referral or referral rate?

Abstract – Could the authors provide more data to the Abstract? I drafted an Abstract for consideration:

Background: Communication and language disorders are common conditions that emerge early and negatively impact quality of life across the life course. Early identification may be facilitated using a validated screening tool such as the Infant Toddler Checklist (ITC). We introduced the ITC at the 18-month visit at child health services in a Swedish county. Using the RE-AIM implementation framework, this study assessed the implementation of the ITC according to five key dimensions: reach, effectiveness, adoption, implementation, and maintenance.

Methods: This was an observational study using medical records at child health services as the data source. Data were collected from children who visited at 17-22 months. We calculated ITC completion rate (reach) and use at each site (adoption). We compared rates of referral to speech and language therapy (effectiveness) before and after implementation of the ITC using Odds Ratio (OR) and 95% Confidence Intervals (CI). We described actions to facilitate implementation and maintenance of ITC screening over time.

Results: The sample included 2633 children with a mean age of ** months, 1717 in the pre-implementation group and 916 in the post-implementation group. The ITC screen positive rate was 14%. The

	overall screening rate was 93% (reach) which increased from 80% initially to 94% at the end of the 2 year period (maintenance). All centres used the ITC (adoption). Of children who had reached at least 30 months (n=2163), referral rate was ***% pre-implementation versus ***% post-implementation (OR 2.07, 95% CI 1.34,3.14) (effectiveness). Implementation strategies included training sessions, collaboration, written and automatic procedures and modifications to the medical records system. Conclusion: The implementation of the ITC was associated with high reach, higher referral rate, complete adoption, and sustained maintenance over time. Title: an alternative title for consideration is: Implementation of the Infant-Toddler Checklist in Swedish child health services at 18 months: an observational study
--	---

VERSION 1 – AUTHOR RESPONSE

Referee 1

Research Questions:

Consider reframing the research questions as specific objectives or aims that correspond with the key dimensions of the RE-AIM framework.

Our response: We apologise if we do not understand the comment by the reviewer correctly. To us, it seems to ask for tying each research question to a RE-AIM dimension. This is already specified at the end of each question, within parentheses. We hope that this is suffice to guide the reader.

Methods:

Include that the ITC is completed by a parent or caregiver.

Our response: We have included the information.

Wetherby and Prizant (the ITC developers) refer to three composite scores and a total score, the composite score terminology is recommended instead of subscales (please see the reference below).
- Reference: Wetherby, Amy & Prizant, Barry. (2001). Communication and Symbolic Behavior Scales Developmental Profile Infant/Toddler Checklist.

https://www.researchgate.net/publication/252234671_Communication_and_Symbolic_Behavior_Scales_Developmental_Profile_InfantToddler_Checklist

Our response: We have rephrased the sentence and added the reference as suggested.

In line 113, instead of the use of the word “fail”, indicate what the cut-off value is (the 10th percentile).

Our response: Thank you for this important comment. We have changed the wording throughout the manuscript, and clarified on the 10th percentile as cut-off (page 5, line 114).

It may be beneficial to include “child age at screening, sex, screening method and results, referral and age at referral to SLT and centre” (referenced in lines 118 & 119) in a table of participant characteristics in the Results (as Table 1).

Our response: Good suggestion. We have added some of this information in Table 1 and the flow chart.

It may be useful to emphasize that informed parent consent was obtained as part of ethics.
Our response: Thank you for asking for clarification. As this study was based on registry data, no parental consent was collected. This has been clarified in the methods section (page 6. Line 132.)

Under study population (line 129) instead of subsample, this seems to be the total sample and a subsample of this was used to determine effectiveness as everyone in the intervention group had not yet reached 30 months of age. It would be beneficial to update the text to refer to the total sample (line 129) followed by the subsample noted on line 134.

Our response: We have taken the reviewer's suggestions into consideration and rephrased the sentences regarding the two samples, (page 6+7. 141-151). Further, we have chosen to redefine the subsample, now including children from 25 months or with early referral, to ease the reader.

Please include a rationale for using 25 months as the cut-point for early referral. The above Wetherby and Prizant (2001) paper may be referenced for this.

Our response: Thank you for this suggestion, this is now amended (page 7. 148-151).

Please provide more details around the type of natural experiment used.

Our response: Thank you for asking for clarification. After considering the comments by reviewer 2, we believe that we somewhat incorrectly described the design as a natural experiment, as the intervention was a planned one and, thus, no "natural" randomisation has occurred. Instead, the study is now described as an observational study, which we think is more accurate.

- Throughout the paper, it is recommended to use the concern terminology used by Wetherby and Prizant rather than "fail".

Our response: Thank you for this comment! We have changed the terminology throughout the manuscript and now use concern where applicable.

Fisher's exact test is typically used to address small cell counts. Please include a brief rationale for the use of Yates' continuity correction.

Our response: Thank you for this comment! The Yate's correction is generally applied to smaller sample sizes, or where a more conservative estimation is called for. We agree with the reviewer that this is not necessary for this study, and have rerun the chi-squared analysis without continuity correction. While this did not change the results in any meaningful way, we think that this choice of analysis is more suitable.

Please clarify the methods used to address the research questions related to the AIM domains.

Our response: Thank you for this suggestion, we have now specified the methods used to assess each AIM domain.

It would be beneficial to include the version of R and SPSS used for this analysis.

Our response: Software versions and references have now been added.

- It would be beneficial to include the alpha level (e.g., 0.05) and whether statistical tests were 2-sided in the methods section.

Our response: Thank you! We apologise for not having the alpha level for the Chi-squared tests included in the methods section. This has now been amended. Regarding one- or two-sided, it is seldomly stated for the Chi-squared test. While it can be viewed as a one-sided test, we do not believe that it is necessary to state.

Results:

- Lines 146 & 147 (Figure 1): Include a title for Figure 1; rephrase the research questions as objectives (as indicated above); for Reach, include the ratio as well as the percentage; and for Effectiveness, include the confidence interval.

Our response: Thank you, the figure (now renamed Table 2) has been revised to address the comments raised by the reviewer.

- Where possible, it may be beneficial to display study results detailed in the text in tables and figures as well.

Our response: We agree with re reviewer, and now have results displayed in tables as well as body text.

Under Effectiveness:

Please clarify whether the 18 children with scores in the concern range on two occasions had speech composite scores in the concern range.

AND

Please clarify whether the following population was captured in this study: children with a speech composite score of concern, who were reassessed after three months and referred to a specialist if the concern remained.

Our response: Good point! We have now clarified this, please see page 8, 175-178.

For the children whose parents expressed no concern, please comment on whether this was based on the ITCs open question on parental concern regarding child development.

There seems to be a slight calculation error: Of the 75 children with scores in the concern range who were not referred early to SLT, only 74 are accounted for in the text.

Our response: Thank you for your keen eyes! Indeed, there was a mistake in the text, where the number of children with parents who had no concern or declined referral should be 10. This has now been corrected.

Discussion:

In lines 221 & 222, reference is made to children who "screened positive with the ITC". A positive ITC screen should be explained in the sections above.

Our response: We have altered the sentence based on your previous suggestion.

Referee 2

Comments to the Author

Thank you for the opportunity to review this manuscript which was of great interest since our research group has also been studying the ITC at the 18 month visit in Canadian primary care settings (reference 23). We agree with the authors that the ITC is a promising screening tool for early detection of children with a range of developmental concerns, including language delay, global developmental delay and autism spectrum disorder. Therefore, I am very enthusiastic about this manuscript. Due to my interest, I read the manuscript carefully and my review is lengthy. However, I think the authors will be able to respond easily to my comments. My most substantial suggestion is to add Tables and Figures to provide greater detail and clarity to the findings. Below are some line-by-line comments.

Our response: Thank you very much for a thorough and competent review of our manuscript! We value the comments and suggestions highly and have responded to all concerns point-by-point below.

Line 68-70 However, screening for language disorder is not recommended for children younger than four.^{11 12} In contrast, early screening is recommended for autism, where language or communication difficulties can be the first symptom.^{11 13} – For readers not familiar with the details of these reviews/recommendations this may be unclear. My interpretation of the current recommendations is that evidence for the effectiveness of screening is weak and screening tools demonstrate poor accuracy. The reviews by Jullien (references 11 and 13) are important, but it is notable that the sources (eg USPSTF, NICE) do not include the most current literature. The authors of the present article may want to modify these sentences.

Our response: We agree on the shortcomings in the two reviews of Jullien, and therefore, we have decided to delete the two sentences.

Line 78 One method for identifying children... - The authors may want to emphasize 'screening', for example: One screening instrument for identifying children...

Our response: We have altered the sentence according to your recommendation.

Line 80-82 When the ITC was used in primary care children with developmental delay were identified with high specificity, high negative predictive value and low false positive rate.²³ – Reference 23 is from our research group. May I suggest a modification to this sentence: When used at the 18 month visit in Canadian primary care, the ITC had high specificity (92%) and negative predictive value (96%), and a low sensitivity (31%) and false positive rate (8%) for a developmental diagnosis at 3–5 years.²³

Our response: Thank you for this suggestion! We have rephrased the sentence.

Line 82-84 ITC used at the 18-month visits in a Swedish CHS showed a sensitivity of 0.85 and specificity 0.59 which increased to 0.88 and 0.63, respectively when the ITC was combined with the nurse's assessment.²⁴ – I have reviewed reference 24 from the same authors. Readers may wonder why the sensitivity and specificity from the two studies (references 23 and 24) are so different. This could be elaborated on in the Discussion section. In reference 24, the authors state in limitations: "Not all children with a positive screen were referred and were therefore not assessed". In our study, criterion measures were available for all children with positive or negative screening. This may account for the differences in study findings.

Our response: Thank you for this comment. The paragraph (line 258–269) has now been edited.

Line 86-91 Setting – I think this paragraph can be moved to the METHODS section.

Our response: We have moved the paragraph according to your recommendation.

Line 92 Implementation strategies – as this paragraph is part of the INTRODUCTION section, I don't think this subheading is needed and could be removed.

Our response: We have deleted the subheadings

Line 102-109 Research questions – This paragraph is well written, clear and much appreciated. The term for each of the 5 dimensions in brackets at the end is very helpful.

Line 106 3. Was the ITC adopted equally across all centres? (Adoption) – Am I correct in understanding that there were 3 centres? Could the number of centres be included here?

Our response: Correct, there were three centres in the region. We have now added this number to the text.

Line 104 2. Did the implementation of the ITC lead to earlier referrals...The authors collect child sex and report on differences in early referrals in Results. Would they consider adding this to research question #2?

Our response: Thank you for this comment! A supplementary question regarding sex differences has been added to research question 2, according to the reviewer's comment. (page 5. Line 101-102).

Line 110 METHODS – the authors could consider adding as the first subheading: Study Design. On Line 136 the authors describe their study as a 'natural experiment' and provide reference 31. I think they could consider identifying their study design as an observational study design using data collected in medical records as the data source.

Our response: Thank you for this important remark. We have changed the description of study type and agree that observational study better fits as description of study design, since there was no "natural" randomisation but rather a planned intervention at a specific time-point.

Line 110 METHODS – After Study Design, the authors could consider a subheading for Setting and Population. As currently written, I found these sections (Setting lines 86-91 and Population lines 128-135) somewhat confusing. For example, the Setting section notes that ITC was introduced in January 2019; but the Population section notes that usual care was received 2016-2020 and ITC was received 2020-2022. Would the authors consider merging these 2 sections and clarify the setting and population?

Our response: Thank you for pointing out this need for clarification. We have rewritten the sections. (page 6, line 134-145).

Line 128-137 Study population – this section is very helpful. The second paragraph describes the study population for research question #2 (effectiveness). Can the authors clarify if the first paragraph describes the study population for research questions #1, 3-5? The word 'subsample' is used – does this mean that this was a subsample of the total sample or is it more accurate to write that these research questions were answered using the total sample? Would the authors consider modifying the first two sentences such as: "To answer the different research questions, two samples were used. The first sample addressed research questions 1,3-5 and included all children..."

Our response: Thank you very much for this insightful comment! We have taken the reviewer's suggestions into consideration and rephrased the sentences regarding the two samples, and have also specified the research questions that are examined with the different samples.

Line 114-115 If only the speech scale indicates concern, the child should be reassessed with the ITC after three months.¹⁶ – Is it correct that if there is concern on the speech composite on two occasions, then the child should be referred for SLT? If so, could this be added?

Our response: This has been added to the manuscript

Line 136 As this study was based on register data, no involvement of patients were possible – Does the journal require a specific subheading “Patient and public involvement”? By using the term ‘register data’ are the authors referring to data collected from medical records? If so, perhaps they can use this terminology (medical records) to be consistent with the terminology used earlier in the manuscript.

Our response: This has been changed to medical records to be consistent with the terms, as you suggested.

Line 138-144 Analysis – The authors describe the analysis plan for research questions #1 and #2. However, I don't think they described the analysis plan for research questions #3-5. Could this be added?

Our response: Thank you for this suggestion! We have now added the analysis plans for assessing the RE-AIM dimensions adoption, implementation and maintenance to the manuscript (page 7. 160-162).

Line 139 and 140 Reach was explored...The effectiveness was explored... - Some methodologists/statisticians prefer to use the word ‘explore’ in specific circumstances. Would the authors consider using the word ‘examined’ or ‘assessed’ rather than ‘explored’?

Our response: Thank you, the word explored is now changed to assessed or examined throughout the manuscript.

Line 142 Effect size was measured through odds ratio – Were 95% confidence intervals calculated? Could these be added?

Our response: Thank you, we have added this to the method.

Line 142 Gender differences... - In Data Collection the authors used the term ‘sex’ which may be more appropriate on Line 142 as well as Figure 1 and Line 204.

Our response: We have changed gender to sex.

Line 145 RESULTS – is it possible for the authors to create a Flow Diagram Figure as per STROBE?

Our response: Thank you for recommending creating a flow chart, we think this makes it easier for the reader to follow. This chart is now included as Fig. 1.

Line 145 RESULTS – Could the authors create a Table 1 Participant Characteristics at screening (n=2633) according to two groups (usual care or ITC) including child age, sex, results of screening (concern for social, speech, and symbolic, and total score), referral (yes/no), age at referral to SLT. These are the variables described in Methods-Data Collection section.

Our response: Thank you for your recommendation. We have added a flowchart as your previous recommendation. We therefore choose not to add a table of participants.

Line 146 Figure 1 – this is a very helpful summary. Could this be labeled a Table rather than a Figure?

Our response: We have changed the label to Table 1 as suggested

Line 152 Effectiveness – this important section would be strengthened with an accompanying Flow Diagram.

Our response: Hopefully, the flowchart of participants strengthens the effectiveness section.

Line 153-154 ... of which 51 children failed on speech – could the authors add the %, ie “of which 51 (42.5%) children failed on speech

Our response: The percentage is added to the manuscript as suggested.

Line 154 ...and thus would be rescreened with the ITC three months later – to clarify, were all 51 rescreened three months later? Or were some missed? These data could be added to a Flow Diagram for clarity.

Our response: As previously described we have added a flowchart.

Line 154-155 For 18 children, screening with the ITC had been performed on two occasions, failing on both occasions – to clarify, is this 18 of 51 (35%) who failed on speech? Does this number (18 of 51) refer to the number who were screened twice (ie 33 of 51 did not receive a second screening)? Or does this mean that all 51 received a second screening and 18 of the 51 (35%) failed both times and were eligible for referral? A Flow Diagram would help clarify.

AND

Lines 155-160 – These sentences are valuable but the interpretation would be strengthened by a Flow Diagram and including %.

Our response: Thank you for this recommendation. We have added a flowchart, but ended up not including percentages as we found it somewhat difficult to interpret.

Line 161 Analysing the subsample of children aged at least 30 months... - this section is very important. Could the authors create a Table 2 of the subsample of children aged at least 30 months (n=2163) according to usual care (n=1717) or ITC (n=446) with similar variables to Table 1 and a column for OR and 95% CI?

Our response: Thank you for this suggestion. A new table has been created, including variables of importance.

In the abstract, the authors present the mean referral age in the usual care vs ITC group; however, this data is not included in the manuscript.

Our response: We apologise for having included an older version of the abstract, where analyses included mean referral age assessment. This has now been removed from the abstract and replaced with odds ratio.

Lines 164-166 Sex differences in early referrals could be reported in Table 2 with %, OR and 95% CI.

Our response: We have now added the results on sex differences to the manuscript.

Line 180 Maintenance – this section would be strengthened with a Figure showing screening rates over time. Methods of assessing and analyzing maintenance were not described in Methods and could be added.

Our response: Thank you for this suggestion, we have added a figure displaying the screening rates over time.

Line 190 DISCUSSION – could the authors summarize their findings in the opening paragraph with a sentence for each of the 5 research questions?

Our response: Good suggestion! The opening paragraph of the discussion has been revised.

Line 200 There was a large difference in referrals between girls and boys using both methods... - could the authors specify the direction and magnitude of the differences (eg two times higher for boys)? And specify the meaning of 'both methods' (ie usual care or ITC screening).

Our response: We have now added the results on sex differences to the manuscript, and added this to the discussion.

Line 201 Language and communication difficulties have been presumed to be more common in boys than in girls – Using the word 'presumed' suggests that there is no literature on this. Can the authors clarify whether there is research evidence to support this? Our research group is also interested in factors associated with positive ITC. We recently reported: The aOR for a positive ITC for male compared with female sex was 2.15 (95% CI, 1.63-2.83; P < .001). (J Pediatr 2024;264:113769).

Our response: Thank you for this suggestion. We have added references to support our statement and the suggested reference regarding ITC.

Line 222 While this was mainly explained through best practice... - can the authors clarify the meaning of this? Are they referring to children who screen positive on the speech composite on first screening? I was unclear in Results section how many of these children underwent a second screening and of these how many screened positive again and were subsequently referred to SLT.

Our response: We have added this to the flowchart.

Line 228 ...as the specificity of the ITC is somewhat low when used at 18 months.²⁴ This is a relatively common challenge in screening at younger ages – This is a very interesting interpretation according to the authors previous research (reference 24). In contrast, our research group has found screening at 18 months in a low risk primary care population to have high specificity (92%) and negative predictive value (96%), and a low sensitivity (31%) and false positive rate (8%) for later developmental outcomes (reference 23 as well as Academic Pediatrics 2023;23:322–328). We have interpreted the findings of low sensitivity from our ITC studies as well as studies of other screening tools at younger ages as highlighting the importance of ongoing developmental surveillance and screening beyond 18 months.

Our response: Thank you for this comment, we have rephrased the section. (page 12, line 264-270).

Conclusion - I note that for effectiveness, the authors state 'earlier referrals', however a comparison of mean age in the two groups is not included in the Methods or Results (but is noted in the Abstract). Was the primary outcome of effectiveness age at referral or referral rate?

Our response: Thank you for pointing out this mistake in the conclusion. The sentence now reads "higher rates of early referrals".

Abstract – Could the authors provide more data to the Abstract? I drafted an Abstract for consideration:

Our response: Thank you for your suggestion and for providing a draft for a new abstract. The abstract is now updated according to suggestions by the reviewer.

VERSION 2 – REVIEW

REVIEWER	Dr. Patricia Parkin Hospital for Sick Children, Paediatrics
REVIEW RETURNED	29-Feb-2024
GENERAL COMMENTS	My thanks to the authors for thoughtfully address all the comments. I have no substantive comments at this time. There are some minor issues to be addressed by the authors or editorial team, eg typographical errors, need for table legends. Thank you.

VERSION 2 – AUTHOR RESPONSE

None